# The global impact of non-alcoholic fatty liver disease (including cirrhosis) in the elderly from 1990 to 2021 and future projections of disease burden

**Siying Huang, Weitao Hu, Taiyong Fang** [ORCID]*

Department of Gastroenterology, The Second Affiliated Hospital of Fujian Medical University, Quanzhou, China

* doctfangty@163.com

## Abstract

### Background

Nonalcoholic fatty liver disease (NAFLD) is a metabolic disorder characterized by hepatic steatosis and inflammation in individuals with no significant alcohol consumption history. Predominantly affecting middle-aged and elderly populations, particularly those with obesity or metabolic syndrome, this condition represents a spectrum ranging from benign fatty accumulation to progressive liver damage. In advanced stages, NAFLD may progress to cirrhosis and hepatocellular carcinoma. This study systematically examines the global incidence patterns and epidemiological characteristics of NAFLD in older adults(>60 years), while establishing predictive models for its future disease burden.

### Methods

Data on NAFLD in the Elderly(>60 years), from 1990 to 2021, was obtained from the Global Burden of Disease (GBD) study, encompassing 204 countries and territories. This dataset includes incidence rates of NAFLD. The Joinpoint regression model was utilized to detect turning points in the epidemiological trends of NAFLD, and decomposition analysis was performed to analyze the factors influencing these trends. To evaluate potential health disparities related to NAFLD, the Slope Index and Concentration Index were calculated. Additionally, the Norpred and Bayesian age-period-cohort (BAPC) models were employed to forecast future incidence rates of NAFLD.

### Results

In 1990, the global NAFLD incidence in the elderly was 2819125(3972309 ± 1807520), with an ASIR of 568.46(803.33 ± 364.10). The global NAFLD prevalence in the elderly was 132549345(166820867 ± 102941502), with an ASPR of

**Data availability statement:** The datasets analyzed during the current study are provided in the Appendix and can also be accessed from the Global Burden of Disease (GBD) database at https://ghdx.healthdata.org/gbd-2021.

**Funding:** The author(s) received no specific funding for this work.

**Competing interests:** This study utilized publicly available, anonymized data from the Global Burden of Disease (GBD) database. No additional ethical approval or informed consent was required. The authors declare no competing interests."

**Abbreviations:** ASIR, Age-standardized incidence rate; ASPR, Age-standardized prevalence rate; CI, Confidence interval; YLDs, Years Lived with Disability; EAPC, Estimated annual percentage change; GBD, Global Burden of Disease; SDI, Socio-Demographic Index; NAFLD, Nonalcoholic fatty liver disease; MASLD, Metabolic dysfunction-associated steatotic liver disease; MAFLD, Metabolic-associated fatty liver disease

27284.94(34321.92±21200.69). The global NAFLD deaths in the elderly were 27864(45975±15898), with an age-standardized death rate of 6.20(10.18±3.55). The global NAFLD DALYs in the elderly were 559945(931920±319045), with an age-standardized DALYs rate of 116.78(193.49±66.73). In 2021, the global NAFLD incidence in the elderly was 7,012,128 (9,896,736±4,480,162), with an ASIR of 636.90 (900.15±406.76). The global NAFLD prevalence in the elderly was 366,363,498 (454,385,769±287,891,088), with an ASPR of 33,576.22 (41,647.44±26,372.13). The global NAFLD deaths in the elderly were 63,313 (99,891±37,267), with an age-standardized death rate of 5.95 (9.39±3.50). The global NAFLD DALYs in the elderly were 1,238,927 (1,973,042±729,228), with an age-standardized DALYs rate of 113.95 (181.50±66.91). From 1990 to 2021, the AAPC of ASIR for NAFLD in the elderly globally was 0.37(0.36 to 0.38), with a p-value < 0.05. The AAPC of ASPR for NAFLD in the elderly globally was 0.67(0.65 to 0.68), with a p-value < 0.05. The AAPC of age-standardized deaths rate for NAFLD in the elderly globally was −0.13(−0.16 to −0.1), with a p-value < 0.05. The AAPC of age-standardized DALYs rate for NAFLD in the elderly globally was −0.05(−0.07 to −0.02), with a p-value < 0.05. The decomposition analysis results indicate that population growth is the primary driver of increased disease burden in older NAFLD patients. It is expected that in the future, the disease burden of NAFLD in elderly people worldwide will continue to rise.

## Conclusions

Over the past three decades, the annual age-standardized incidence rate and total number of cases of NAFLD, including cirrhosis, have increased among the elderly population, irrespective of gender. This upward trend is consistent across all SDI regions. Furthermore, future projections indicate that both the annual age-standardized incidence rate and the case numbers of NAFLD, including cirrhosis, in the elderly are likely to continue rising.

## Introduction

Nonalcoholic fatty liver disease (NAFLD), also referred to as metabolic-associated fatty liver disease (MAFLD) or metabolic dysfunction-associated steatotic liver disease (MASLD), is a clinical-pathological syndrome characterized by hepatic steatosis in patients without excessive alcohol use and other established etiologies of liver injury [1]. It is closely associated with insulin resistance and genetic susceptibility, representing an acquired metabolic stress-related liver injury [2]. NAFLD includes simple fatty liver (SFL), nonalcoholic steatohepatitis (NASH), and related cirrhosis [3]. Autoimmune liver diseases have also been suggested to have a potential association with NAFLD [4]. Additionally, patients with celiac disease initiating a gluten-free diet have a higher risk of developing NAFLD [5]. Patients with NAFLD have a higher risk

of sudden cardiac death [6]. With the global rise of obesity and associated metabolic syndrome, NAFLD has become a significant cause of chronic liver disease in many countries [7].

As people age, they face an increased risk of various metabolic diseases, with NAFLD being one of the most common [8]. This is particularly true for individuals aged 65 and older. Aging is associated with decreased metabolic function, reduced physical activity, and changes in dietary habits, all of which contribute to age-related increases in NAFLD risk [9]. Obesity is a major risk factor for NAFLD [10]. Abdominal obesity, especially the accumulation of visceral fat, remains a significant contributor to NAFLD in older patients [11]. Additionally, the risk of diabetes increases with age, and diabetes is closely linked to the development of NAFLD [12]. Diabetic patients tend to have more pronounced hepatic fat accumulation and a higher risk of liver dysfunction [13]. Hypertension and dyslipidemia, other components of metabolic syndrome, are also common in the elderly and contribute significantly to the risk of NAFLD [14]. Furthermore, many elderly individuals require long-term medication for chronic diseases, and certain drugs (such as corticosteroids and certain antibiotics) may increase the risk of developing NAFLD [15].

## Methods

### Materials

The 2021 Global Burden of Disease Study (GBD 2021) provides a comprehensive estimate of the incidence rate of nonalcoholic fatty liver disease (NAFLD), including cirrhosis. These estimates are derived from data collected during the GBD 2021 study and can be accessed through the GBD Results Viewer at https://vizhub.healthdata.org/gbd-results/. The most recent access to this database was on December 2, 2024. The data sources for NAFLD, including cirrhosis, include a wide range of records from various countries, such as hospitalization data, emergency room visits, insurance claims, surveys, and vital registration systems [16].

The research utilized anonymized data compiled by the Institute for Health Metrics and Evaluation (IHME) at the University of Washington. The study protocol, including the waiver of informed consent, was reviewed and approved by the Institutional Review Board of the University of Washington [17]. The 2021 Global Burden of Disease Study (GBD 2021) provides an extensive analysis of 369 diseases and injuries, along with 87 related risk factors, meticulously evaluated across 204 countries and territories [18]. The GBD research team has provided a comprehensive explanation of their methodological approach these health conditions [19].

To enable comparative analysis, the Global Burden of Disease project categorizes nations based on their Socio-Demographic Index (SDI). The SDI is calculated as the geometric mean of three key indicators: per capita income, average educational attainment for individuals aged 15 and older, and the fertility rate among women under 25. This index divides countries and regions into five distinct SDI tiers: high SDI (>0.81), upper-middle SDI (0.70–0.81), middle SDI (0.61–0.69), lower-middle SDI (0.46–0.60), and low SDI (<0.46). By incorporating these social and economic factors, this classification framework highlights both the commonalities and disparities in epidemiological patterns across different nations and regions [20].

### Statistical analysis

**Description of disease burden.** To better characterize the disease burden of NAFLD in the elderly, we calculated the incidence, prevalence, deaths, DALYs (Disability-Adjusted Life Years), and corresponding age-standardized rates of NAFLD among populations aged ≥60 years in 2021 across the globe, SDI regions, and 204 countries and territories, using age-stratified disease burden data and population demographics.

**Average annual percent change.** The Average Annual Percentage Change (AAPC) is a composite metric that summarizes the direction and magnitude of trends over a specified time period, providing a single value to represent the average yearly variation during that interval [21]. For all AAPC calculations, we used Joinpoint software, a tool developed by the Surveillance Research Program at the National Cancer Institute [22]. Decomposition analysis was used to assess

the impact of age, population growth, and epidemiological trends on the incidence of non-alcoholic fatty liver disease, including cirrhosis [23]. To analyze trends in the etiology of NAFLD in the elderly, it is essential to understand the changing composition of its underlying causes.

**Decomposition analysis.** To further understand the drivers behind recent changes in NAFLD incidence trends among older patients, we used decomposition analysis to examine the roles of population growth, population aging trends, and disease prevalence trends in altering the NAFLD disease burden [23].

**Health inequality.** Health inequality refers to disparities between different groups in terms of health status, access to healthcare, and factors influencing health outcomes. In this study, the author uses the slope index of absolute health inequity and the concentration index of relative health inequity to assess whether health inequities exist among countries across various Socio-Demographic Index (SDI) regions concerning the incidence rate of non-alcoholic fatty liver disease, including cirrhosis [24]. The slope index is calculated using both the ordinary linear regression model (LM) and the robust regression model (iterative weighted linear regression, RLM). Unlike ordinary linear regression, robust regression is better suited for data with high heteroscedasticity, as it effectively downweights outliers. The choice between these models depends on the results of the Non-constant Variance Test (ncvTest): if the p-value exceeds 0.05, indicating no heteroscedasticity, the LM model is used; if the p-value is less than 0.05, suggesting the presence of heteroscedasticity, the RLM model is applied. If multiple groups show inconsistent ncvTest results, the outcomes of the RLM model may be influenced. A positive slope index indicates an upward skew in the distribution of health indicators between groups, meaning a higher proportion of groups have superior health metrics. Conversely, a negative slope index suggests a downward skew, with fewer groups exhibiting favorable health indicators. The magnitude of the slope index directly reflects the severity of inequality in health indicator distributions between groups. Similarly, the concentration index assesses the concentration or dispersion of health indicators across groups. A positive concentration index indicates a concentrated distribution, with a larger proportion of groups experiencing better health outcomes. A negative value, on the other hand, signifies a more dispersed distribution, suggesting fewer groups with higher health indicators. The magnitude of the concentration index highlights the degree of concentration or dispersion. To statistically compare global variations in the incidence rate of non-alcoholic fatty liver disease, including cirrhosis, across all countries from 1990 to 2021, a Z-test was employed, providing a robust assessment of the significance of observed differences.

**Prediction of long-term disease burden.** The Norpred model and Bayesian age-period-cohort models(BAPC) were employed to forecast the future incidence. Both the Norpred models are grounded in the Age-Period-Cohort (APC) model [25]. The theoretical foundation of this model posits that incidence or mortality is influenced by age structure and population size. The projected demographic data is derived from GBD 2017.

All analyses in this study were performed using R software (R Core Team, version 4.3.3, Vienna, Austria), with a p-value of less than 0.05 considered statistically significant.

## Ethics declarations

The study utilized de-identified data, which was compiled by the Institute for Health Metrics and Assessment (IHME) at the University of Washington. The University of Washington Institutional Review Board reviewed and approved the informed consent waiver for this study [26].

## Results

### Global burden analysis from 1990 to 2021

In 1990, the global NAFLD incidence in the elderly was 2,819,125 (3,972,309 ± 1,807,520), with an ASIR of 568.46 (803.33 ± 364.10). The global NAFLD prevalence was 132,549,345 (166,820,867 ± 102,941,502), with an ASPR of 27,284.94 (34,321.92 ± 21,200.69). Deaths were 27,864 (45,975 ± 15,898), with an age-standardized death rate of 6.20 (10.18 ± 3.55). DALYs were 559,945 (931,920 ± 319,045), with an age-standardized rate of 116.78 (193.49 ± 66.73). In

2021, the global NAFLD incidence rose to 7,012,128 (9,896,736±4,480,162), with an ASIR of 636.90 (900.15±406.76). Prevalence reached 366,363,498 (454,385,769±287,891,088), with an ASPR of 33,576.22 (41,647.44±26,372.13). Deaths increased to 63,313 (99,891±37,267), with a death rate of 5.95 (9.39±3.50). DALYs totaled 1,238,927 (1,973,042±729,228), with a rate of 113.95 (181.50±66.91). S1–S4 Tables present the incidence, prevalence, deaths, and DALYs of NAFLD among older patients in 204 countries and territories. Fig 1 shows the incidence, prevalence, deaths, and DALYs of NAFLD among older patients across different SDI regions.

## Average annual percent change

From 1990 to 2021, the AAPC of ASIR for NAFLD in the elderly globally was 0.37(0.36 to 0.38), with a p-value<0.05. In high SDI regions, the AAPC was 0.43(0.43 to 0.44), p-value<0.05. In high-middle SDI regions, the AAPC was 0.36(0.34 to 0.37), p-value<0.05. In middle SDI regions, the AAPC was 0.24(0.23 to 0.25), p-value<0.05. In low-middle SDI regions, the AAPC was 0.22(0.22 to 0.23), p-value<0.05. In low SDI regions, the AAPC was 0.19(0.18 to 0.19), p-value<0.05. From 1990 to 2021, the AAPC of ASPR for NAFLD in the elderly globally was 0.67(0.65 to 0.68), with a p-value<0.05. In high SDI regions, the AAPC was 0.77(0.77 to 0.78), p-value<0.05. In high-middle SDI regions, the AAPC was 0.7(0.68 to 0.72), p-value<0.05. In middle SDI regions, the AAPC was 0.53(0.51 to 0.55), p-value<0.05. In low-middle SDI regions, the AAPC was 0.44(0.43 to 0.44), p-value<0.05. In low SDI regions, the AAPC was 0.31(0.31 to 0.31), p-value<0.05. From 1990 to 2021, the AAPC of age-standardized deaths rate for NAFLD in the elderly globally was −0.13(−0.16 to −0.1), with a p-value<0.05. In high SDI regions, the AAPC was −0.32(−0.35 to −0.28), p-value<0.05. In high-middle SDI regions, the AAPC was −0.77(−0.82 to −0.71), p-value<0.05. In middle SDI regions, the AAPC was 0.3(0.27 to 0.33), p-value<0.05. In low-middle SDI regions, the AAPC was 0.07(0.02 to 0.12), p-value<0.05. In low SDI regions, the AAPC was −0.31(−0.33 to −0.28), p-value<0.05. From 1990 to 2021, the AAPC of age-standardized DALYs rate for NAFLD in the elderly globally was −0.05(−0.07 to −0.02), with a p-value<0.05. In high SDI regions, the AAPC was −0.26(−0.29 to −0.23), p-value<0.05. In high-middle SDI regions, the AAPC was −0.77(−0.82 to −0.71), p-value<0.05. In middle SDI regions, the AAPC was 0.41(0.37 to 0.45), p-value<0.05. In low-middle SDI regions, the AAPC was 0.22(0.19 to 0.26), p-value<0.05. In low SDI regions, the AAPC was −0.4(−0.42 to −0.38), p-value<0.05. Fig 2 illustrates the epidemiological

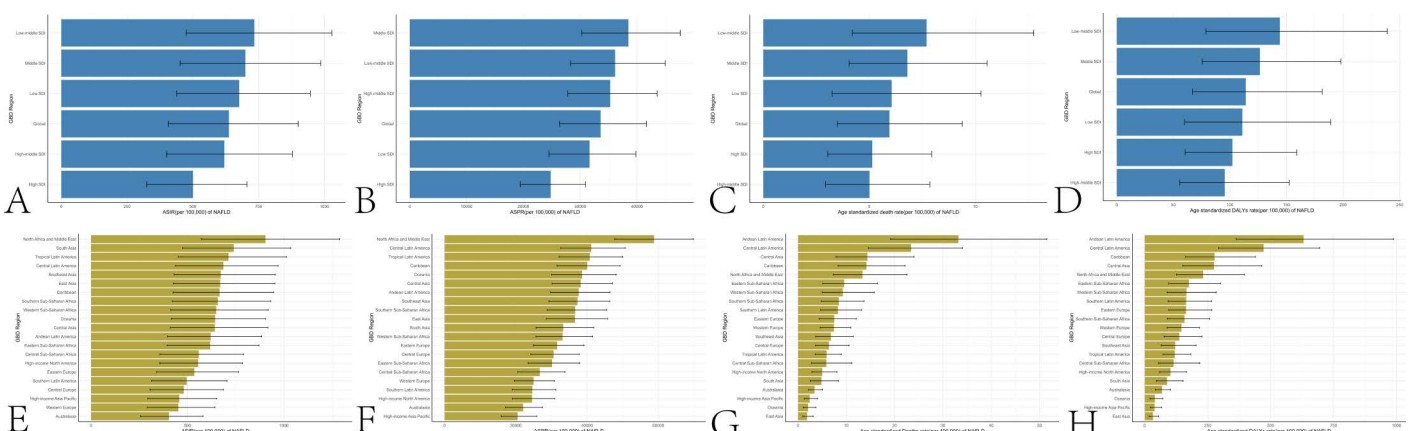

**Fig 1. Global disease burden of NAFLD in the elderly across SDI regions.** A: ASIR of NAFLD in the elderly in high-to-low SDI regions. B: ASPR of NAFLD in the elderly across 5 SDI regions. C: Age standardized Death rate of NAFLD in the elderly across 5 SDI regions. D: Age standardized dALYs rate of NAFLD in the elderly across 5 SDI regions. E: ASIR of NAFLD in the elderly across 21 SDI regions. F: ASPR of NAFLD in the elderly across 21 SDI regions. H: Age standardized death rate of NAFLD in the elderly across 21 SDI regions. H: Age standardized DALYs rate of NAFLD in the elderly across 21 SDI regions.

trends of NAFLD in the elderly from 1990 to 2021, presenting the ASIR, ASPR, age-standardized deaths rate and age-standardized DALYs rate.

## Decomposition analysis

The decomposition analysis results indicate that population growth is the primary driver of increased disease burden in older NAFLD patients. These results are presented in Table 1 and Fig 3.

## Health inequality

The visualization results of the slope index and concentration index of the incidence rate of NAFLD among elderly patients globally are depicted in Fig 4. In the Z-test, between 1990 and 2021, the Z-value of the global comparison of the incidence rate of Nonalcoholic fatty liver disease including cirrhosis is 2.21, with a P-value is 0.027.

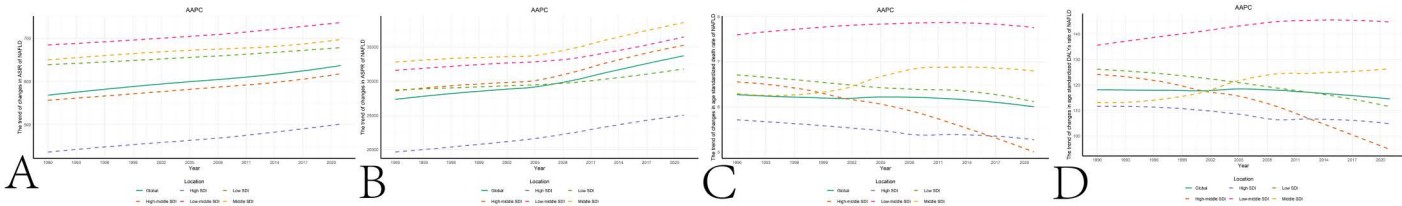

**Fig 2. Trends in ASIR, ASPR and of age-standardized YLDs rate from 1990 to 2023 based on Joinpoint regression analysis.** A: Trend of ASIR. B: Trend of ASPR. C: Trend of age-standardized deaths rate. D: Trend of age-standardized DALYs rate.

**Table 1. Decomposition analysis results of incidence rate of Nonalcoholic fatty liver disease including cirrhosis in the world and different SDI regions from 1990 to 2021.**

| Measure | Gender | Overll difference | Aging | Population | Epidemiological change |
|---|---|---|---|---|---|
| Incidence | Both | 4193002.88 | −66954.22(−1.6%) | 3697036.39(88.17%) | 562920.71(13.43%) |
| | Male | 1811394.78 | −38629.68(−2.13%) | 1686547.45(93.11%) | 163477.01(9.02%) |
| | Female | 2381608.1 | −30729.01(−1.29%) | 2007166.86(84.28%) | 405170.24(17.01%) |
| Prevalence | Both | 233814153.4 | 211127.81 (0.09%) | 183839487.62 (78.63%) | 49763538 (21.28%) |
| | Male | 110627801.5 | 656107.75 (0.59%) | 89423003.61 (80.83%) | 20548690.19 (18.57%) |
| | Female | 123186351.9 | −338635.67 (−0.27%) | 94603910.33 (76.8%) | 28921077.23 (23.48%) |
| Deaths | Both | 35449.29 | 2304.39 (6.5%) | 34878.56 (98.39%) | −1733.66 (−4.89%) |
| | Male | 15644.98 | 965.54 (6.17%) | 14618.89 (93.44%) | 60.55 (0.39%) |
| | Female | 19804.32 | 1232.08 (6.22%) | 20117.37 (101.58%) | −1545.14 (−7.8%) |
| DALYs | Both | 678981.99 | 7105.95 (1.05%) | 691457.3 (101.84%) | −19581.26 (−2.88%) |
| | Male | 312254.56 | 970.37 (0.31%) | 305141.23 (97.72%) | 6142.96 (1.97%) |
| | Female | 366727.44 | 4021.48 (1.1%) | 384616.28 (104.88%) | −21910.32 (−5.97%) |

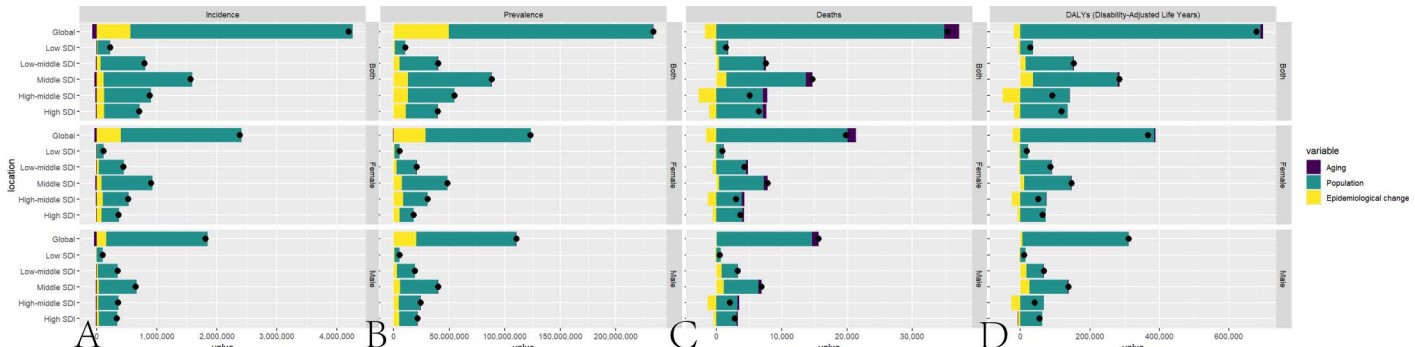

**Fig 3. Results of decomposition analysis.** A: Decomposition analysis of NAFLD incidence among older patients from 1990 to 2021. B: Decomposition analysis of NAFLD prevalence among older patients from 1990 to 2021. C: Decomposition analysis of NAFLD-related deaths among older patients from 1990 to 2021. D: Decomposition analysis of NAFLD DALYs among older patients from 1990 to 2021.

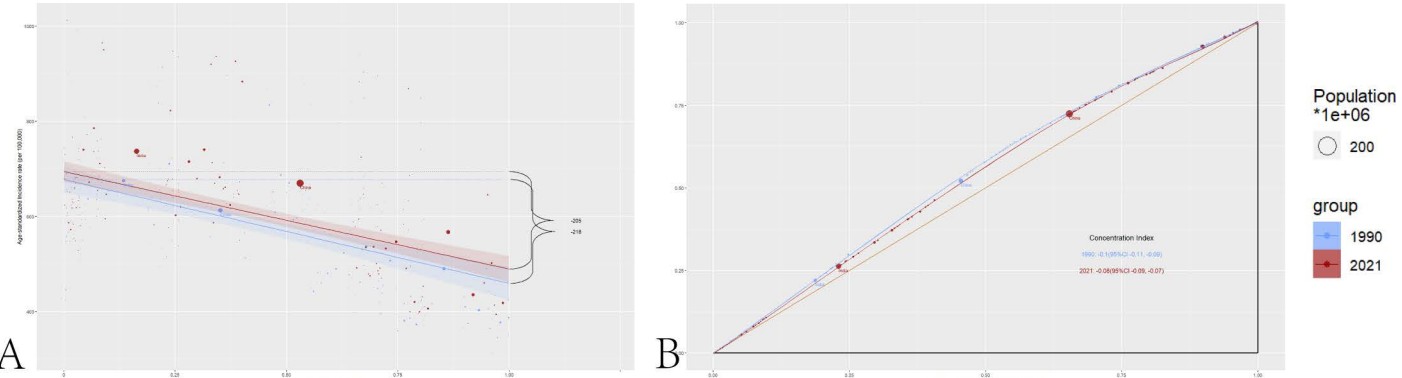

**Fig 4. The visualization results of the slope index and concentration index globally and across different SDI regions.** A:The slope index of the incidence rate of NAFLD among elderly patients globally. B:The concentration index of the incidence rate of NAFLD among elderly patients globally.

## Prediction of long-term disease burden

The Nordpred model has been employed to forecast the incidence in the future. Predictions indicate that the standardized incidence rate of NAFLD among elderly in the year 2046 is estimated to be 655.29 cases per 100,000 population. This figure comprises an estimate of 597.85 cases for males and 706.02 cases for females. The total estimated incidence of NAFLD among elderly by the year 2046 is projected to be 12772348 cases, with 5430895 cases expected among males and 7341453 cases among females. The BAPC model has been employed to forecast the incidence in the future. Predictions indicate that the standardized incidence rate of NAFLD among elderly in the year 2046 is estimated to be 785.24 cases per 100,000 population. This figure comprises an estimate of 728.36 cases for males and 774.25 cases for females. The total estimated incidence of NAFLD among elderly by the year 2046 is projected to be 15192786 cases, with 6885372 cases expected among males and 8307414 cases among females. The results of predicting the future incidence rate of NAFLD among elderly are presented in Fig 5.

## Discussion

Based on our research, we found that compared to 1990, the incidence, prevalence, deaths, and DALYs of NAFLD in the elderly population in 2021 had more than doubled, indicating that the disease burden of NAFLD in older adults remains

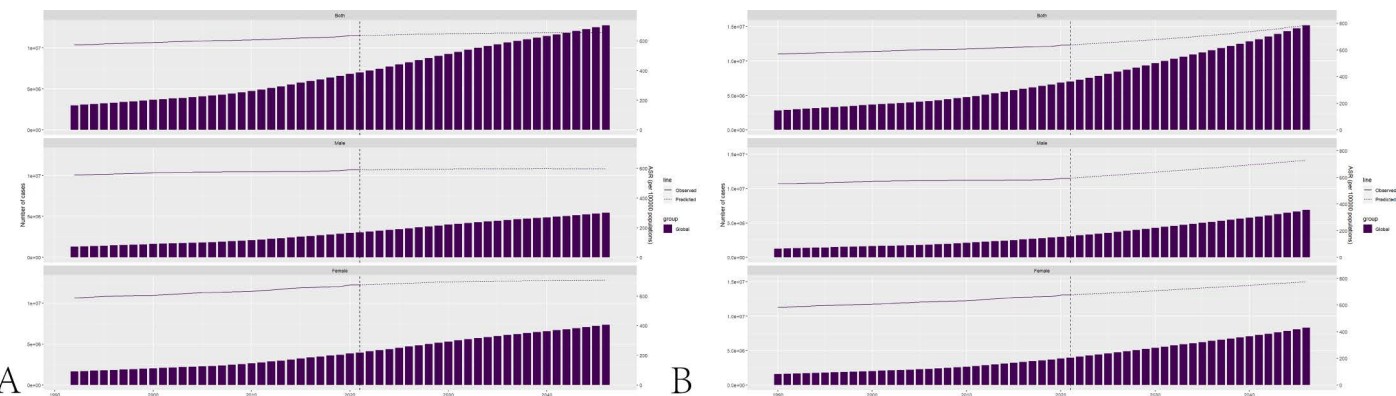

**Fig 5. The results of predicting the future incidence rate.** A: Projection of future NAFLD incidence in older adults based on the Nordpred model. B: Projection of future NAFLD incidence in older adults based on the BAPC model.

substantial. Further analysis revealed that both the ASIR and ASPR of NAFLD in the elderly increased in 2021 compared to 1990. Fortunately, the age-standardized death rate and age-standardized DALYs rate demonstrated a declining trend over this period.

To analyze the causes of these findings, we conducted a decomposition analysis. The results suggest that the global population growth, particularly the increase in the elderly population, has contributed to the rising disease burden of NAFLD in older patients. Additionally, the dramatic rise of NAFLD risk factors such as obesity and diabetes in the elderly population has accelerated the progression of NAFLD in this group [27]. The widespread use of imaging techniques and biomarker technologies has also led to the identification of more asymptomatic NAFLD cases among older adults [28]. These findings highlight the continued necessity for global NAFLD prevention in the elderly population. Specific measures include avoiding unhealthy lifestyles such as high-calorie diets and sedentary behaviors [29]. Furthermore, interventions targeting metabolic disorders like insulin resistance should be further promoted [30].

It is encouraging that despite the rising disease burden of NAFLD, the age-standardized death rate and age-standardized DALYs rate of NAFLD in 2021 showed a decline compared to 1990. This improvement is associated with the widespread adoption of NAFLD screening programs, which enabled earlier detection and management of patients through liver function monitoring and imaging techniques before the progression to fibrosis or cirrhosis [31]. Additionally, measures to control metabolic syndromes—such as using GLP-1 receptor agonists to manage diabetes and statins to regulate blood lipids—have contributed to slowing NAFLD progression and reducing NAFLD-related mortality [32].

Our study used Joinpoint regression to explore the epidemiological trends of NAFLD disease burden in the elderly across global and different SDI regions. We found that high-SDI regions had the largest increases in ASIR and ASPR, likely due to the combined effects of unhealthy lifestyles and metabolic disorders [33]. The slowest growth rates in low-SDI regions may be linked to limited diagnostic capabilities leading to underreporting rather than lower actual risk. High- and high-middle SDI regions showed significant declines in age-standardized mortality and DALYs rates for elderly NAFLD, while middle- and low-middle SDI regions experienced clear increases in both mortality and DALYs. Low-SDI regions exhibited a paradoxical trend of declining mortality alongside rising incidence, potentially due to underdiagnosis or indirect effects from viral hepatitis interventions. These results underscore the need for SDI-specific NAFLD control strategies: high-SDI regions should intensify metabolic syndrome management to curb rising ASIR/ASPR [34]; middle- and low-middle SDI regions must prioritize obesity/diabetes control and enhance primary care screening with portable ultrasound and low-cost biomarkers [35]; low-SDI regions need improved diagnostic awareness and surveillance systems to differentiate NAFLD from infectious liver diseases [36].

To provide reference for further preventive measures, we employed two predictive models to forecast the incidence of NAFLD, including cirrhosis, in the elderly population [37]. It is projected that by 2046, the ASIR of NAFLD in elderly individuals would reach 655.29–785.24, with the total number of cases ranging from 12,772,348–15,192,786.

Fortunately, an analysis of health inequities among elderly patients with non-alcoholic fatty liver disease (NAFLD), including cirrhosis, in 1990 and 2021 revealed improvements. The results indicate that both the skewness index and the concentration index in 2021 showed improvement compared to 1990, with the difference in concentration indices between the two years being statistically significant. This suggests that health inequities among elderly patients with NAFLD, including cirrhosis, have improved globally in recent years. Globally, many countries and regions have gradually strengthened public health policies, with an increasing focus on the prevention and treatment of NAFLD in the elderly population [38]. These efforts include initiatives aimed at preventing related conditions such as obesity, hyperglycemia, and hyperlipidemia [39]. Such policies have provided the elderly with better opportunities for disease prevention and treatment, thus reducing health disparities between different social groups [40]. In addition, the widespread promotion of health education has helped improve awareness of NAFLD among the elderly. More and more elderly individuals are understanding the impact of lifestyle on liver health and are adopting healthier dietary habits and regular physical activity [41]. With the improvement of the global economy, an increasing number of elderly individuals, particularly in economically disadvantaged areas, are gaining access to better healthcare services [42]. The health status of impoverished and marginalized groups has significantly improved, thereby reducing health inequities among elderly NAFLD patients [43].

This study is based on data from the GBD research, which provides high-quality estimates, but also has certain unavoidable limitations. First, the GBD study categorizes data by country and region, without including ethnic demographic information, which may overlook the impact of race on the incidence of NAFLD, including cirrhosis. As a result, analyzing and comparing global trends and changes in the incidence of NAFLD, as well as examining differences between SDI levels and age groups within each racial group, presents challenges. Second, early data or data from countries and regions with lower levels of development may be less accurate. Additionally, the extended 30-year time span is susceptible to various uncontrollable factors that could influence the prediction of age-standardized disease rates. Additionally, the risk factors for NAFLD were not described in the study. These limitations should be taken into account when interpreting the results of this study.

## Conclusion

Over the past 30 years, both the annual age-standardized incidence rate and the number of cases of non-alcoholic fatty liver disease, including cirrhosis, have shown an increasing trend among the elderly population, regardless of gender. This upward trend has been observed across all SDI regions. Furthermore, in the future, both the annual age-standardized incidence rate and the number of cases of non-alcoholic fatty liver disease, including cirrhosis, in the elderly population are expected to continue rising.

## Supporting information

**S1 Table. Incidence of NAFLD among eldly in all countries and regions in 1990 and 2021.**
(PDF)

**S2 Table. Prevalence of NAFLD among eldly in all countries and regions in 1990 and 2021.**
(PDF)

**S3 Table. Deaths of NAFLD among eldly in all countries and regions in 1990 and 2021.**
(PDF)

**S4 Table. DALYs of NAFLD among eldly in all countries and regions in 1990 and 2021.**
(PDF)

## Author contributions

**Data curation:** Siying Huang.

**Writing – original draft:** Siying Huang, Weitao Hu.

**Writing – review & editing:** Taiyong Fang.

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
