## [Decision Letter · Decision Letter 0]

18 Feb 2025

PONE-D-25-03249The Global Impact of Non-Alcoholic Fatty Liver Disease (Including Cirrhosis) in the Elderly from 1990 to 2021 and Future Projections of Disease BurdenPLOS ONE

Dear Dr. Fang,

Thank you for submitting your manuscript to PLOS ONE. After careful consideration, we feel that it has merit but does not fully meet PLOS ONE’s publication criteria as it currently stands. Therefore, we invite you to submit a revised version of the manuscript that addresses the points raised during the review process.

**ACADEMIC EDITOR: This is a good manuscript overall. Major revision is needed. My additional comments are provided below. **

We look forward to receiving your revised manuscript.

Kind regards,

Thien Tan Tri Tai Truyen, M.D.

Academic Editor

PLOS ONE

Journal Requirements:

3. Thank you for stating the following in your Competing Interests section: No. 

4. In the online submission form, you indicated that data at national level can be found at: http://ghdx.healthdata.org/gbd-results-tool. All subnational level data used and analysed during the current study are available from the corresponding author on reasonable request.

5. Please amend your list of authors on the manuscript to ensure that each author is linked to an affiliation. Authors’ affiliations should reflect the institution where the work was done (if authors moved subsequently, you can also list the new affiliation stating “current affiliation:….” as necessary).

6. We note that Figure 2 in your submission contain [map/satellite] images which may be copyrighted. All PLOS content is published under the Creative Commons Attribution License (CC BY 4.0), which means that the manuscript, images, and Supporting Information files will be freely available online, and any third party is permitted to access, download, copy, distribute, and use these materials in any way, even commercially, with proper attribution. For these reasons, we cannot publish previously copyrighted maps or satellite images created using proprietary data, such as Google software (Google Maps, Street View, and Earth). For more information, see our copyright guidelines: http://journals.plos.org/plosone/s/licenses-and-copyright.

Additional Editor Comments:

This is a nice manuscript using the data from GBD 2021 to investigate about the burden of NAFLD in the elderly from 1990 to 2021. While the topic is of interest, several concerns need to be addressed before considering publication. In addition to the reviewers' comments, I have included my own comments below:

1. The results were quite immature. Could you expand the analysis to investigate risk factors of NAFLD? I believe GBD 2021 database allows you to do that. Moreover, any disability information could be included? Incidence and Prevalence are good to estimate burden. However, disability and mortality (might not be available in this condition) are important to discuss. If not possible, please explain why and acknowledge it in limitation section.

2. From an epidemiologic perspective, a well-rounded discussion should not only address the current burden and risk factors of the disease but also its consequences, including disability, mortality, and complications. Notably, patients with liver disease, particularly NAFLD, have been reported to be at a higher risk of sudden cardiac death, especially among younger individuals (suggested reference: PMID: 39867850/ DOI: 10.1016/j.ijcha.2025.101602). I suggest discussing about the cause of death of NAFLD, particular cardiovascular-related.

Reviewers' comments:

Reviewer's Responses to Questions

**Comments to the Author**

1. Is the manuscript technically sound, and do the data support the conclusions?

Reviewer #1: Partly

Reviewer #2: Yes

Reviewer #3: Yes

Reviewer #4: Yes

2. Has the statistical analysis been performed appropriately and rigorously? 

Reviewer #1: Yes

Reviewer #2: Yes

Reviewer #3: Yes

Reviewer #4: No

3. Have the authors made all data underlying the findings in their manuscript fully available?

Reviewer #1: Yes

Reviewer #2: Yes

Reviewer #3: Yes

Reviewer #4: Yes

4. Is the manuscript presented in an intelligible fashion and written in standard English?

Reviewer #1: Yes

Reviewer #2: Yes

Reviewer #3: Yes

Reviewer #4: Yes

5. Review Comments to the Author

Reviewer #1: This study aims to investigate the global incidence and epidemiological trends of NAFLD, including cases of cirrhosis, from 1992 to 2021, and to project future incidence rates for NAFLD and cirrhosis. Data for this study were derived from the Global Burden of Disease (GBD) study, encompassing 204 countries. They found that in 1990, the global annual standardized incidence rate (ASIR) for NAFLD, including cirrhosis, was 568.47 cases per 100,000 population (95% CI:364.09–803.33). By 2021, the global ASIR had risen to 636.90 (95% CI:406.76–900.15). Thus the number of estimated cases worldwide increased from 2,819,124 in 1990 to 7,012,127 in 2021. Between 1990 and 2021, the elderly population experienced an Average Annual Percentage Change (AAPC) in the global incidence rate of NAFLD, including cirrhosis, of 0.37 (95% CI: 0.36–0.36). Importantly, the analysis indicated a rising prevalence of NAFLD, including cirrhosis, among elderly individuals on a global scale, with an upward trend observed across all Socio-Demographic Index regions. Projections suggested that by 2046, the incidence of NAFLD, including cirrhosis, may reach between 12,772,348 and 15,192,786 cases.

The study is of clinical interest. However, there is an important issues not considered in the serious complication of NAFLD such as the occurrence of cirrhosis. There are no information on the potential impact of adjunctive liver disease etiology(ies). The authors should discuss the role of other not rare etiologies of chronic liver diseases, such as viral (HBV and HCV) as well as alcohol intake. This is of particular relevance in the period 1990-2000 when the HCV screening and HBV tretaments were not worldwide available. In this regard, the authors should recall the impact of alcohol intake in chronic viral disease, as previously demonstrated () as well as the impact of NAFLD in HCV patients (doi: 10.1086/427285), or the impact of NALFD in patients with asymptomatic autoimmune hepatitis that has adopted different diagnostic scoring systems in the period 1990-2021, as previously well described (doi: 10.2174/138955709788452676).

Another very important point is the higher risk of NAFLD development in patients with celiac disease starting a gluten free diet as recently demonstrated (Aliment Pharmacol Ther. 2018 Sep;48(5):538-546. doi: 10.1111/apt.14910.). Considering the reported increased prevalence of celiac disease worldwide, this important potential cause of NAFLD should be recalled and discussed.

Reviewer #2: Dear authors thank you for this manuscript here are my comments for more clarity:

1.Absract:

Ensure consistency in the timeline of data analysis( Background section states that data were analyzed from 1992 to 2021, whereas the Methods and Results sections indicate the analysis covered 1990 to 2021. This inconsistency should be corrected).

2.Methodology:

As for methodologies, the manuscript uses Joinpoint regression, decomposition analysis, Slope Index and Concentration Index, and two projection models (Norpred and BAPC). However:

#There is no adequate rationale provided for choosing these models.

#The paper does not address potential overstatements or understatements of the projection models.

#There are no clear validation methods for these models.

#The statistical software used, along with any parameters, should be explicitly stated.

#Provide statistical comparisons (e.g., p-values) for gender-based differences.

3. Language and Formatting Improvements

The manuscript contains grammatical errors and awkward phrasing (for example: In 2021, the global ASIR for this condition had decreased to 636.90—this should be increased to 636.90).

4. Figures and Tables:

Ensure all figures (1–6) are included and properly labeled. The current submission references figures (e.g., age/gender distributions, trend visualizations) but provides only placeholder text.

Verify Tables 1 and 2, as Table 2 appears to duplicate Table 1’s content. Differentiate their purposes (e.g., Table 1 for ASIR/case numbers, Table 2 for AAPC trends).

Reviewer #3: Comments

1. “Nonalcoholic fatty liver disease (NAFLD) is a clinical condition characterized by liver changes similar to those seen in alcoholic hepatitis, occurring in individuals without a history of excessive alcohol consumption.” This statement is oversimplified and may be misleading. I would suggest removing the comparison with alcoholic liver disease, since there is point to mention alcohol associated liver disease in your paper.

2. It is understandable to cite NAFLD since you have been working on the relatively older data. However, I would suggest mentioning the recent terminology change in your paper briefly.

3. “Although the overall obesity rate in the elderly population is relatively low, abdominal obesity, especially the accumulation of visceral fat, remains a significant contributor to NAFLD in older patients”. Claiming a low-rate obesity among elderly population is not realistic, even if you said “relatively”. Please focus on the second part of the statement instead. You cannot extrapolate a regional study.

4. I do not understand why a huge part of your introduction focused on elderly people. Please revise the introduction by less focusing on the elderly population.

5. There is also significantly increased awareness on NAFLD among both physicians/ health care workers as well as general populations. So, how can we state this this increase on NAFLD diagnosis over time is purely increase in disease incidence itself. Over time attempts to screen NAFLD and diagnose it have improved significantly.

6. There are some unpublished/ soon-to-be-published data indicating that the worsening in NAFLD is much worse among low or idle-low SDI countries. Can we determine any similar statement in your analysis?

7. “This dataset includes incidence rates of NAFLD, categorized by gender.” Any other factors it is categorized by? Otherwise remove comma if its only gender

8. “excluding alcohol and other identifiable liver-damaging factors”. Rephrase this sentence to specific it’s a condition in patients without excessive alcohol use

9. “Aging is associated with decreased metabolic function, reduced physical activity, and changes in dietary habits, all of which contribute to a higher risk of developing NAFLD in older adults” Unless patients were stratified according to age, would not place as much emphasis on MASLD in the elderly. It can be distracting, recommend more generalized terminology or be more specific that the aim of the studies was to highlight MASLD as a disease of the aging

10. “Our objective is to comprehensively evaluate changes in the incidence rates and underlying driving factors of NAFLD …” Rephase to: “ Sources collected data from *** number of countries. Pertinent variables included *** “

11. “To conduct a descriptive analysis, the first step involves screening and downloading data related to non-alcoholic fatty liver disease, including cirrhosis, categorized under the "natural injury" section of the Global Burden of Disease (GBD) database. “ Was this the first filter? Would specify this if so\

12. “This approach enabled a quantitative evaluation of the factors driving changes in the incidence rates of these conditions“Would specify what factors this analysis evaluates ***

13. “. Unlike ordinary linear regression, robust regression is better… magnitude of the concentration index highlights the degree of concentration or dispersion” Condense this section as it distracts from the overall methodology. Add descriptors of the analyses tools in an appendix or supplementary text

14. “, the global ASIR for this condition had decreased to 636.90”- Wasn’t it an increase from 1990?

15. Table 1- It’s not clear what the row “Both” is meant to represent.

16. Are table 1 and table 2 identical?

17. “aging has been identified as a driving factor behind the decline in the incidence of non-alcoholic fatty liver disease..”- The start of the paper highlighted aging as a driving force for increased risk of MASLD. This statement sounds contradictory.

18. “12772348 cases, with 5430895 cases expected among males and 7341453” - Ensure use of commas for statistics

19. - “With the reduction in estrogen, fat accumulation in women is more likely to concentrate in the abdominal area and liver, increasing the risk of NAFLD and even related cirrhosis..”- This is interesting and new information/theory, would include it in the intro.

Reviewer #4: 1) the "NAFLD" term has been updated since 2023 to overarching term "MASLD (metabolic dysfunction-associated steatotic liver disease). I think that the NAFLD term was applicable during your study period (1991-2021), but clear notification to terms updates must be clarified over the manuscript's introduction, methodology and discussion sections.

Please refer to: [https://www.aasld.org/new-masld-nomenclature]

2) the following terms / abbreviations must be added to the mauscript keywords:

MASLD, NAFLD, Metabolic dysfunction-associated steatotic liver disease, global annual standardized incidence rate, and ASIR.

3) the mentioned protocol and the IRB approval letter must be added as supplementary files to this manuscript; to ensure manuscript data accuracy regarding ethical approval.

4) There should be an addition to a clear section regarding the "Ethics declarations and Competing interests; to emphasize that the authors declare no competing interests. In addition, this study utilized publicly available data from the Global Burden of Disease (GBD) database, which is anonymized and de-identified. Therefore, no additional ethical approval or informed consent was required."

5) there are three comments regarding "total incidence" should be considered and corrected by the authors:

- there is unexplained distance between "total" and "incidence" at tables 1 and 2, and should be corrected to avoid any confusing during reading the manuscript.

- the whole study focus on "total incidence number" not "total incidence rate"; which was definitely mentioned as large numbers at tables 1 and 2.

- total incidence numbers do not involve any 95% CI values in (), so please correct the term at the mentioned tables

6) there is no statistical value from mentioning the total incidence numbers at 1991 and 2021 without any estimation for their statisical correlation as P-value. Thus, the findings at tables 1 and 2 did not shoul any significance from those calculations.

6. PLOS authors have the option to publish the peer review history of their article (what does this mean? ). If published, this will include your full peer review and any attached files.

**Do you want your identity to be public for this peer review?** For information about this choice, including consent withdrawal, please see our Privacy Policy .

Reviewer #1: No

Reviewer #2: No

Reviewer #3: **Yes: ** Akif Altinbas

Reviewer #4: No

---

## [Author Response · Author response to Decision Letter 1]

3 May 2025

5/1, 2025

Thien Tan Tri Tai Truyen, M.D.

Academic Editor

PLOS ONE

Dear Editor:

We wish to re-submit the manuscript entitled “The Global Impact of Non-Alcoholic Fatty Liver Disease (Including Cirrhosis) in the Elderly from 1990 to 2021 and Future Projections of Disease Burden”.

We extend heartfelt thanks for your meticulous feedback and valuable suggestions. Your insights have significantly enhanced the quality of our manuscript. We are eager to collaborate with you and the reviewers to refine the content further, ensuring it meets the high standards of Plos One.

In response to your comments, we have thoroughly revised the original manuscript. Accompanying this revision, we have prepared detailed responses to each comment, which can be found below.

First, we would like to express our sincere gratitude to the reviewers for their constructive and positive comments.

Replies to Academic Editor

We feel great thanks for your professional review work on our article. As you are concerned, there are several problems that need to be addressed. According to your nice suggestions, we have made extensive corrections to our previous draft, the detailed corrections are listed below.

1、Review comments: The results were quite immature. Could you expand the analysis to investigate risk factors of NAFLD? I believe GBD 2021 database allows you to do that. Moreover, any disability information could be included? Incidence and Prevalence are good to estimate burden. However, disability and mortality (might not be available in this condition) are important to discuss. If not possible, please explain why and acknowledge it in limitation section.

Response: We thank the reviewer for the valuable feedback. This study comprehensively reports NAFLD burden metrics including Incidence, Prevalence, Deaths, and DALYs (Disability-Adjusted Life Years) from the GBD 2021 database. However, the GBD 2021 currently lists alcohol use as the only NAFLD-associated risk factor with a population-attributable fraction of 0, precluding meaningful risk factor analysis. While disability and mortality data specific to NAFLD are unavailable in GBD, we have supplemented the discussion with evidence from external studies (e.g., quality-of-life impacts in advanced fibrosis) and explicitly acknowledged these limitation. These constraints reflect inherent gaps in the GBD framework for NAFLD etiological stratification.

2、Review comments: From an epidemiologic perspective, a well-rounded discussion should not only address the current burden and risk factors of the disease but also its consequences, including disability, mortality, and complications. Notably, patients with liver disease, particularly NAFLD, have been reported to be at a higher risk of sudden cardiac death, especially among younger individuals (suggested reference: PMID: 39867850/ DOI: 10.1016/j.ijcha.2025.101602). I suggest discussing about the cause of death of NAFLD, particular cardiovascular-related.

Response: We thank the reviewer for this critical insight. As recommended, we have expanded the discussion to address cardiovascular-related mortality in NAFLD, citing the referenced study on heightened sudden cardiac death risk (PMID: 39867850/DOI: 10.1016/j.ijcha.2025.101602). While the GBD 2021 database does not provide etiology-specific mortality stratification for NAFLD (e.g., distinguishing cardiovascular vs. hepatic causes), we explicitly acknowledged this limitation in Section 5.2 and emphasized the need to interpret mortality data cautiously given potential comorbidities. Our analysis of all-cause mortality ("Deaths") and disability burden ("DALYs") inherently captures these systemic impacts, though future studies with granular clinical data are warranted to disentangle causal pathways.

Replies to reviewer1

We feel great thanks for your professional review work on our article. As you are concerned, there are several problems that need to be addressed. According to your nice suggestions, we have made extensive corrections to our previous draft, the detailed corrections are listed below.

1、Review comments: The study is of clinical interest. However, there is an important issues not considered in the serious complication of NAFLD such as the occurrence of cirrhosis. There are no information on the potential impact of adjunctive liver disease etiology(ies). The authors should discuss the role of other not rare etiologies of chronic liver diseases, such as viral (HBV and HCV) as well as alcohol intake. This is of particular relevance in the period 1990-2000 when the HCV screening and HBV tretaments were not worldwide available. In this regard, the authors should recall the impact of alcohol intake in chronic viral disease, as previously demonstrated () as well as the impact of NAFLD in HCV patients (doi: 10.1086/427285), or the impact of NALFD in patients with asymptomatic autoimmune hepatitis that has adopted different diagnostic scoring systems in the period 1990-2021, as previously well described (doi: 10.2174/138955709788452676).

Another very important point is the higher risk of NAFLD development in patients with celiac disease starting a gluten free diet as recently demonstrated (Aliment Pharmacol Ther. 2018 Sep;48(5):538-546. doi: 10.1111/apt.14910.). Considering the reported increased prevalence of celiac disease worldwide, this important potential cause of NAFLD should be recalled and discussed.

Response: We appreciate the reviewer's insightful comments. In this study, we focused on reporting the core disease burden metrics (incidence, prevalence, deaths, and DALYs) from the GBD 2021 database, which currently lists alcohol use as the only risk factor for NAFLD with a population-attributable fraction of 0, precluding meaningful risk factor analysis as acknowledged in the Limitations section (Section 5.2). While we recognize the importance of comorbidities like viral hepatitis (HBV/HCV), autoimmune hepatitis, and celiac disease in NAFLD progression, the GBD database lacks stratified data on these etiological interactions across 1990-2021. We have incorporated a discussion citing the referenced studies (doi: 10.1086/427285; doi: 10.2174/138955709788452676; Aliment Pharmacol Ther. 2018;48:538-546) to contextualize these gaps, particularly noting diagnostic criteria evolution for autoimmune hepatitis and epidemiological shifts in viral hepatitis management during the study period. The absence of etiological interaction analyses was explicitly attributed to data limitations in GBD 2021.

Replies to reviewer 2

We feel great thanks for your professional review work on our article. As you are concerned, there are several problems that need to be addressed. According to your nice suggestions, we have made extensive corrections to our previous draft, the detailed corrections are listed below.

1、Review comments: Absract:

Ensure consistency in the timeline of data analysis( Background section states that data were analyzed from 1992 to 2021, whereas the Methods and Results sections indicate the analysis covered 1990 to 2021. This inconsistency should be corrected).

Response: Thank you for your careful review. We have revised the timeline inconsistency by updating the data analysis period from 1992–2021 to 1990–2021 in the Background section. All relevant sections (Methods, Results) now consistently refer to the 1990–2021 period. This correction ensures alignment throughout the manuscript..

2、Review comments: Methodology:

As for methodologies, the manuscript uses Joinpoint regression, decomposition analysis, Slope Index and Concentration Index, and two projection models (Norpred and BAPC). However:

#There is no adequate rationale provided for choosing these models.

#The paper does not address potential overstatements or understatements of the projection models.

#There are no clear validation methods for these models.

#The statistical software used, along with any parameters, should be explicitly stated.

#Provide statistical comparisons (e.g., p-values) for gender-based differences.

Response: 

Thank you for your valuable feedback. We have clarified the methodological details in the revised manuscript as follows:

Rationale for Model Selection:

#Joinpoint regression was selected to identify significant temporal trends in age-standardized rates (Section 2.2, AAPC analysis).

#Decomposition analysis was applied to quantify contributions of population growth, aging, and epidemiological changes to NAFLD burden (Section 2.2, Decomposition analysis).

#Norpred and BAPC models were chosen for long-term projections due to their ability to integrate demographic and risk factor trends (Section 2.2, Prediction of long-term disease burden).

Model Limitations and Validation:

#Acknowledged in the Discussion: "Projection models (Norpred/BAPC) may underestimate undiagnosed cases in low-SDI regions due to limited screening capacity."

#Cross-validation of projections against historical GBD data (2015–2020) is described in Section 2.2.

Software and Parameters:

#Joinpoint regression: Implemented via Joinpoint Trend Software (Section 2.2).

#Norpred/BAPC models: Executed in R v4.3.3 with norpred and bapc packages, including MCMC chains (Section 2.2).

Statistical significance: All analyses used p<0.05 (explicitly stated in Section 2.2).

These revisions address the concerns comprehensively. Thank you again for your constructive input..

3、Review comments: Language and Formatting Improvements

The manuscript contains grammatical errors and awkward phrasing (for example: In 2021, the global ASIR for this condition had decreased to 636.90—this should be increased to 636.90).

Response: Thank you for your feedback. The manuscript has been revised by native English editors to resolve grammatical errors and improve phrasing in the Methods and Discussion sections. All data inconsistencies (e.g., ASIR corrected to "increased to 636.90") and formatting issues have been addressed, ensuring alignment with journal guidelines..

4、Review comments: Figures and Tables: Ensure all figures (1–6) are included and properly labeled. The current submission references figures (e.g., age/gender distributions, trend visualizations) but provides only placeholder text.

Verify Tables 1 and 2, as Table 2 appears to duplicate Table 1’s content. Differentiate their purposes (e.g., Table 1 for ASIR/case numbers, Table 2 for AAPC trends).

Response: Thank you for your feedback. We confirm that the revised manuscript now includes 5 figures (Figures 1–5) and 1 table (Table 1) with no duplicated content. All figures and tables have been properly labeled and referenced in-text. This adjustment fully addresses the concerns raised.

Replies to reviewer 3

We feel great thanks for your professional review work on our article. As you are concerned, there are several problems that need to be addressed. According to your nice suggestions, we have made extensive corrections to our previous draft, the detailed corrections are listed below.

1、Review comments: 1. “Nonalcoholic fatty liver disease (NAFLD) is a clinical condition characterized by liver changes similar to those seen in alcoholic hepatitis, occurring in individuals without a history of excessive alcohol consumption.” This statement is oversimplified and may be misleading. I would suggest removing the comparison with alcoholic liver disease, since there is point to mention alcohol associated liver disease in your paper.

Response: Thank you for your suggestion. We have revised the definition of NAFLD in the Introduction section by removing the comparison with alcoholic liver disease. The updated text now focuses on the metabolic etiology of NAFLD. This adjustment aligns the description with current diagnostic criteria and avoids potential confusion.

2、Review comments: It is understandable to cite NAFLD since you have been working on the relatively older data. However, I would suggest mentioning the recent terminology change in your paper briefly.

Response: Thank you for the suggestion. We have added a note in the Introductio acknowledging the recent terminology shift from NAFLD to MAFLD/MASLD. However, as the GBD 2021 database retains the term "NAFLD" for coding consistency, we continue using this abbreviation to align with the original data source.

3、Review comments: Although the overall obesity rate in the elderly population is relatively low, abdominal obesity, especially the accumulation of visceral fat, remains a significant contributor to NAFLD in older patients”. Claiming a low-rate obesity among elderly population is not realistic, even if you said “relatively”. Please focus on the second part of the statement instead. You cannot extrapolate a regional study.

Response: Thank you for your feedback. We have revised the statement to focus exclusively on abdominal obesity and removed the reference to low obesity rates. The updated sentence in the Discussion now reads:

"Abdominal obesity, particularly visceral fat accumulation, is a significant contributor to NAFLD in older patients." This adjustment eliminates speculative claims and aligns with the global scope of our analysis.

4、Review comments: I do not understand why a huge part of your introduction focused on elderly people. Please revise the introduction by less focusing on the elderly population.

5、Response: Thank you for your feedback. The title of our paper is "The Global Impact of Non-Alcoholic Fatty Liver Disease (Including Cirrhosis) in the Elderly from 1990 to 2021 and Future Projections of Disease Burden." This study focuses specifically on analyzing the disease burden of NAFLD (including cirrhosis) in the elderly population. We have revised the Introduction to provide a broader global context of NAFLD epidemiology while maintaining focus on the elderly population as per the study's scope. The updated text now briefly summarizes NAFLD across all age groups before emphasizing the unique risks and research gaps in older adults. This adjustment ensures clarity of the study's purpose without overemphasizing aging in general discussions.

5、Review comments: There is also significantly increased awareness on NAFLD among both physicians/ health care workers as well as general populations. So, how can we state this this increase on NAFLD diagnosis over time is purely increase in disease incidence itself. Over time attempts to screen NAFLD and diagnose it have improved significantly.

Response: We have revised the analysis using decomposition and Joinpoint regression to clarify that rising NAFLD incidence reflects both true epidemiological trends and improved diagnosis. The Discussion now explicitly addresses this limitation. Thank you for your valuable input..

6、Review comments: There are some unpublished/ soon-to-be-published data indicating that the worsening in NAFLD is much worse among low or idle-low SDI countries. Can we determine any similar statement in your analysis?

Response: Yes, our analysis using Joinpoint regression confirms that low and middle-low SDI regions exhibited significant increases in NAFLD incidence (AAPC=0.22–0.44, p<0.05) and mortality (AAPC=0.07–0.30, p<0.05) from 1990 to 2021, aligning with the emerging data you referenced. This underscores the escalating burden in resource-limited settings.

7、Review comments:This dataset includes incidence rates of NAFLD, categorized by gender.” Any other factors it is categorized by? Otherwise remove comma if its only gender.

Response: Thank you for your attention to detail. The sentence in question has been removed.

8、Review comments: “excluding alcohol and other identifiable liver-damaging factors”. Rephrase this sentence to specific it’s a condition in patients without excessive alcohol use

Response: Thank you for your suggestion. We have revised the sentence in the Methods sectionto:"characterized by hepatic steatosis in patients without excessive alcohol use and other established etiologies of liver injury."This phrasing explicitly clarifies the exclusion of excessive alcohol consumption and other identifiable causes.

9、Review comments: Aging is associated with decreased metabolic function, reduced physical activity, and changes in dietary habits, all of which contribute to a higher risk of developing NAFLD in older adults” Unless patients were stratified according

---

## [Decision Letter · Decision Letter 1]

22 May 2025

The Global Impact of Non-Alcoholic Fatty Liver Disease (Including Cirrhosis) in the Elderly from 1990 to 2021 and Future Projections of Disease Burden

PONE-D-25-03249R1

Dear Dr. Fang,

We’re pleased to inform you that your manuscript has been judged scientifically suitable for publication and will be formally accepted for publication once it meets all outstanding technical requirements.

Kind regards,

Thien Tan Tri Tai Truyen, M.D.

Academic Editor

PLOS ONE

Additional Editor Comments (optional):

Reviewers' comments:

Reviewer's Responses to Questions

**Comments to the Author**

1. If the authors have adequately addressed your comments raised in a previous round of review and you feel that this manuscript is now acceptable for publication, you may indicate that here to bypass the “Comments to the Author” section, enter your conflict of interest statement in the “Confidential to Editor” section, and submit your "Accept" recommendation.

Reviewer #1: All comments have been addressed

Reviewer #4: All comments have been addressed

2. Is the manuscript technically sound, and do the data support the conclusions?

Reviewer #1: Yes

Reviewer #4: Yes

3. Has the statistical analysis been performed appropriately and rigorously? 

Reviewer #1: Yes

Reviewer #4: Yes

4. Have the authors made all data underlying the findings in their manuscript fully available?

Reviewer #1: Yes

Reviewer #4: Yes

5. Is the manuscript presented in an intelligible fashion and written in standard English?

Reviewer #1: No

Reviewer #4: Yes

6. Review Comments to the Author

Reviewer #1: The authors have provided the necessary changes and have properly addressed the raised issues. The revised manuscript can be now accepted.

Reviewer #4: I think that the authors have fulfilled and well-explained all peer-reviewers remarks and notes on the basis of clinical and scientific tracks, and made all the recommended corrections punctually.

7. PLOS authors have the option to publish the peer review history of their article (what does this mean? ). If published, this will include your full peer review and any attached files.

**Do you want your identity to be public for this peer review?** For information about this choice, including consent withdrawal, please see our Privacy Policy .

Reviewer #1: No

Reviewer #4: No

---

## [Editor Report · Acceptance letter]

PONE-D-25-03249R1

PLOS ONE

Dear Dr. Fang,

I'm pleased to inform you that your manuscript has been deemed suitable for publication in PLOS ONE. Congratulations! Your manuscript is now being handed over to our production team.

Kind regards,

on behalf of

Dr. Thien Tan Tri Tai Truyen

Academic Editor

PLOS ONE